

# Screening for drought tolerance in cultivars of the ornamental genus *Tagetes* (Asteraceae)

Raluca Cicevan[1],*, Mohamad Al Hassan[2],*, Adriana F. Sestras[1], Jaime Prohens[3], Oscar Vicente[2], Radu E. Sestras[1] and Monica Boscaiu[4]

[1] Faculty of Horticulture, University of Agricultural Sciences and Veterinary Medicine of Cluj-Napoca, Cluj-Napoca, Romania
[2] Institute of Plant Molecular and Cellular Biology (IBMCP, UPV-CSIC), Departamento de Biotechnologia, Universitat Politècnica de València (UPV), Valencia, Spain
[3] Instituto de Conservación y Mejora de la Agrodiversidad Valenciana, Universitat Politècnica de València, Valencia, Spain
[4] Instituto Agroforestal Mediterráneo, Universitat Politècnica de València, Valencia, Spain
* These authors contributed equally to this work.

Corresponding author
Monica Boscaiu,
Mobosnea@eaf.upv.es

## ABSTRACT

Drought tolerance was evaluated in twelve cultivars of three ornamental *Tagetes* species (*T. patula*, *T. tenuifolia* and *T. erecta*). A stress treatment was performed by completely stopping watering of plants maintained in controlled greenhouse conditions. After three weeks, several plant growth parameters (stem length (SL), fresh weight (FW) and water content (WC)), photosynthetic pigments (chlorophylls and carotenoids (Car)), osmolytes (proline (Pro), glycine betaine (GB) and total soluble sugars (TSS)), an oxidative stress maker (malondialdehyde (MDA)) and antioxidants (total phenolic compounds (TPC) and total flavonoids (TF)) were measured. Considerable differences in the evaluated traits were found among the control and drought-stressed plants. Drought stress generally caused a marked reduction in plant growth and carotenoid pigments, and an increase in soluble solutes and oxidative stress. For most cultivars, proline levels in stressed plants increased between 30 and 70-fold compared to the corresponding controls. According to the different measured parameters, on average *T. erecta* proved to be more tolerant to drought than *T. patula* and *T. tenuifolia*. However, a considerable variation in the tolerance to drought was found within each species. The traits with greater association to drought tolerance as well as the most tolerant cultivars could be clearly identified in a principal components analysis (PCA). Overall, our results indicate that drought tolerant cultivars of *Tagetes* can be identified at early stages using a combination of plant growth and biochemical markers.

## INTRODUCTION

The genus *Tagetes* L. (Asteraceae) includes 53 annual and perennial species (*The Plant List, 2013*) native to the American continent, from SW United States to South America. Several *Tagetes* species, commonly called marigolds, are well-known ornamental and

medicinal plants cultivated throughout the world. When used as ornamentals they are mostly cultivated in flower beds and borders of landscape settings or as cut flowers (*Valdez-Aguilar, Grieve & Poss, 2009*). Regarding their medicinal interest, extracts of leaves or flowers have been used in different affections of skin, kidney or liver (*Giri, Bose & Mishra, 2011*; *Maity et al., 2011*); many species have also an anti-inflammatory action (*Shetty, Harikiran & Fernandes, 2009*; *Shinde et al., 2009*), or are used to reduce hypertension and high levels of cholesterol (*Raghuveer et al., 2011*). Moreover, studies on different taxa of this genus proved their antimicrobial (*Ruddock et al., 2011*), insecticidal (*Hollis, Gonzalez & Walsh, 2012*), and nematicidal (*Kiranmai, Kazim & Ibrahim, 2011*) effects. Their economic importance is increased by their value as melliferous plants and their use in the cosmetic industry, or as a natural dye for textiles, and food products (*Vasudevan, Kashyap & Sharma, 1997*; *Jothi, 2008*).

This study focuses on three *Tagetes* species that are used as bedding ornamentals in gardens and green areas but also as potted plants in terraces or balconies, and occasionally as cut flowers. The French marigold (*T. patula* L.), native from Mexico to Argentina and first reported in Europe from France (*Adams, 2004*), is now cultivated throughout the world. It is well acclimated to Mediterranean regions, where is one of the most common ornamental species. Most of the numerous cultivars which have been developed give plants that are small sized, usually not more than 40 cm tall, much branched, with dark green leaves and small floral heads. *Tagetes erecta* L., commonly known as African marigold—although it originates from Mexico—is also grown in many regions outside its native range. Plants of this species can reach up to 1.5 m or even more in its native area, but the most commonly used cultivars are generally shorter, with a height of 60–80 cm (*Serrato-Cruz, 2004*). Although plants with long stems are optimal for making flower garlands, overgrowth leads to an unattractive plant appearance that hinders their use in pots. For this reason, new commercial varieties of *T. erecta* are usually selected for shorter growth (*Serrato-Cruz, 2004*). The third species under study, *T. tenuifolia* Cav. or signet marigold, with smaller leaves and floral heads, is less known but holds promise for development as a commercially important ornamental crop (*Gilman & Howe, 1999*).

Ornamental *Tagetes* species are common in Mediterranean areas. Climate change has dramatically increased the frequency and extension of drought episodes in the last decades in many areas of the world, with the Mediterranean region being one of the most affected (*Giannakopoulos et al., 2009*). The rapid global warming, besides an increase in mean temperatures worldwide, is causing more frequent, longer and more intense extreme weather phenomena, such as droughts, 'heat waves,' or floods (*IPCC, 2014*). Climatic scenarios predict that by the end of this century, Mediterranean regions will suffer the effects of higher temperatures and reduced precipitation, which will cause desertification in some areas (*Rubio, 2009*).

Mitigation of global warming is a formidable challenge at present, and there is an urgent need of selecting more stress tolerant genotypes of cultivated plants (*Gholinezhad, Darvishzadeh & Bernousi, 2014*). Considering that in the near future, water will be a scarcer and more expensive resource and that irrigation will be more

restrictively used, selection and diversification of stress tolerant cultivars should be a priority for contemporary ornamental horticulture (*Niu, Rodriguez & Wang, 2006*). In addition, on many occasions, potted plants for home or garden decoration suffer from drought stress due to a lack of regular watering by customers. Traditionally, selection of drought tolerant cultivars was done by directly growing plants under water stress (WS) conditions and comparing their growth and reproductive parameters with those measured in plants in the control treatments. Although this approach is reliable and has been successfully applied, it requires a long period of experimentation. Nowadays, faster screening methods are available (*Mantri, Patade & Pang, 2014*). Several of these have been already applied in *Tagetes*, such as testing the capacity of seeds to germinate in conditions of osmotic stress, simulated by PEG (*Cicevan et al., 2015*), or the in vitro selection of drought tolerant clones (*Mohamed, Harris & Henderson, 2000*). A tool of remarkable utility, quick and detectable in very early stages of plant growth is the use of stress biochemical markers. This wide category comprises numerous compounds, easy to be quantified (*Ashraf & Harris, 2004*; *Schiop et al., 2015*), which after a brief exposure to stress suffer a change in their concentration in the plants, either increasing or decreasing with respect to the values registered in the controls. Optimal markers for screening the drought tolerance of crops are those related to degradation of photosynthetic pigments (*Bijanzadeh & Emam, 2010*; *Mafakheri et al., 2010*; *Siddiqui et al., 2015*), or osmolytes that are usually synthesised in conditions of cell dehydration as it occurs when salinity or water stresses are applied (*Sofo et al., 2004*; *Talukdar, 2013*). Drought, as many other abiotic stresses (high salinity, extreme temperatures), results in the enhanced generation of reactive oxygen species (ROS). These secondary metabolites are continuously produced in plants as by-products of aerobic metabolism, but under environmental stress conditions, their amount may largely increase, leading to oxidative stress (*Van Breusegem & Dat, 2006*). Consequently, a general response to abiotic stress in plants is based on the activation of enzymatic and non-enzymatic antioxidant systems; the latter category including many flavonoids and other phenolic compounds (*Apel & Hirt, 2004*). If drought stress is prolonged beyond the tolerance limits of each genotype, ROS production surpasses the limits of scavenging action of the antioxidant systems, leading to the oxidation of the amino acid residues in proteins, the unsaturated fatty acids in cell membranes, and DNA molecules, causing extensive cellular damage and eventually plant death (*Halliwell, 2006*).

The aim of this work was two-fold: first, the identification of marigold tolerant cultivars, based on their relative growth inhibition under severe drought conditions; and, second, to establish which biochemical responses are the best indicators of stress in *Tagetes*. The achievement of this latter objective may allow setting up protocols for quick evaluation of drought tolerance in *Tagetes* on the basis of biochemical markers.

## MATERIALS AND METHODS

### Plant material

The cultivars selected for this study were purchased from local companies in Romania (Agrosem) and Hungary (Kertimag) and comprised five cultivars of *T. patula* ('Bolero,'
'Orange Flame,' 'Orion,' 'Robuszta,' and 'Szinkeverek'), four of *T. tenuifolia* ('Luna Gold,' 'Luna Lemon,' Luna Orange,' and 'Sarga'), and three of *T. erecta* ('Alacsony Citromsarga,' 'Aranysarga,' and 'Cupid Golden Yellow').

## Plant cultivation

All seeds were sown directly into a moistened mixture of peat (50%), perlite (25%) and vermiculite (25%) in 1 L pots (Ø = 11 cm). The substrate was kept moderately moist, using Hoagland's nutritive solution. Three weeks after seedling emergence, a water stress treatment was initiated by completely stopping irrigation, maintaining a control treatment in which plants were watered twice a week with Hoagland's nutritive solution (125 mL per pot). All experiments were conducted in a controlled environment chamber in a greenhouse, under the following conditions: long-day photoperiod (16 h of light obtained by supplementing natural light with artificial light), temperature of 23 °C during the light period and 17 °C during the dark period. Air humidity ranged between 50–80% during the course of the experiment. Three weeks after the initiation of treatments, when water stressed plants were already clearly affected, all material was harvested and the following growth parameters were determined: stem length (SL; cm), fresh weight (FW; g) of the leaves, dry weight of the leaves (DW, g) and water content (WC; %) (*Gil et al., 2014*).

## Photosynthetic pigments

Chlorophyll a, chlorophyll b and total carotenoids were measured following *Lichtenthaler & Wellburn (1983)*. Basically, 100 mg fresh plant material was crushed and extracted with 30 ml 80% ice-cold acetone prior to being vortexed and centrifuged. The supernatant was separated and its absorbance was measured at 663 nm ($A_{663}$), 646 nm ($A_{646}$), and 470 nm ($A_{470}$), using a Cadex model SB038 spectrophotometer (Cadex, Saint-Jean-sur-Richelieu, Quebec, Canada). The concentration of each group of compounds was calculated according to the following equations: Chlorophyll *a* (chl a; $\mu g \cdot ml^{-1}$) = $12.21 \cdot A_{663} - 2.81 \cdot A_{646}$; Chlorophyll *b* (chl b; $\mu g \cdot ml^{-1}$) = $20.13 \cdot A_{646} - 5.03 \cdot A_{663}$; Total chlorophylls ($\mu g \cdot ml^{-1}$) = chl a + chl b; Total carotenoids ($\mu g \cdot ml^{-1}$) = $(1{,}000 \cdot A_{470} - 3.27 \cdot [\text{chl a}] - 104 \cdot [\text{chl b}])/227$. The values were later converted to $mg \cdot g^{-1}$ DW.

## Osmolyte analysis

Proline (Pro) content was determined in 100 mg of fresh leaves by the ninhydrin-acetic acid method of *Bates, Waldren & Teare (1973)*. Pro was extracted in 3% aqueous sulfosalicylic acid, the extract was mixed with acid ninhydrin solution, incubated for 1 h at 95 °C, cooled on ice and then extracted with two volumes of toluene. Absorbance of the organic phase was measured at 520 nm, using toluene as a blank. Pro concentration was expressed as $\mu mol \cdot g^{-1}$ DW.

Glycine betaine (GB) was determined in 100 mg of dried plant material, according to the method described by *Grieve & Grattan (1983)*. The sample was ground with 2 ml of Mili-Q water, and then extracted with 4 ml of 1, 2-dichlorethane; absorbance of the

solution was measured at a wavelength of 365 nm. GB concentration was expressed as $\mu$mol·g$^{-1}$ DW.

Total soluble sugars (TSS) were quantified following *Dubois et al. (1956)*. One hundred mg of dried leaf material was ground and mixed with 3 ml of 80% methanol on a rocker shaker for 24–48 h. Concentrated sulfuric acid and 5% phenol was added to the sample and absorbance was measured at 490 nm. TSS contents were expressed as mg of glucose equivalents g$^{-1}$ DW. All three osmolytes' wavelengths were measured using a Cadex model SB038 spectrophotometer.

## Oxidative stress marker and non-enzymatic antioxidants

Malondialdehyde (MDA), total antioxidant flavonoids (TF), and total phenolic compounds (TPC) were determined in 80% (v/v) methanol extracts of 100 mg of dry plant material.

MDA, a final product of membrane lipid peroxidation and a reliable marker of oxidative stress (*Del Rio, Stewart & Pellegrini, 2005*), was determined as reported by *Hodges et al. (1999)*. Extracts were mixed with 0.5% thiobarbituric acid (TBA) prepared in 20% TCA, (or with 20% TCA without TBA for the controls), and then incubated at 95 °C for 20 min. After stopping the reaction, the supernatant absorbance was measured at 532 nm. The non-specific absorbance at 600 and 440 nm was subtracted and MDA concentration was calculated using the equations described in *Hodges et al. (1999)*.

TF were measured following the method described by *Zhishen, Mengcheng & Jianming (1999)*, mixing the methanol extracts with sodium nitrite, followed by aluminum chloride and sodium hydroxide. Absorbance was measured at 510 nm, and the amount of antioxidant total flavonoids was expressed in equivalents of catechin, used as standard (mg eq. C·g$^{-1}$ DW).

TPC were quantified as described in *Blainski, Lopes & Palazzo de Mello (2013)*, by reaction with the Folin-Ciocalteu reagent. The extracts were mixed with the reagent and sodium bicarbonate and left in the dark for 90 min. Absorbance was recorded at 765 nm, and the results expressed in equivalents of gallic acid, used as standard (mg eq. GA·g$^{-1}$ DW). All wavelengths were measured using a Cadex model SB038 spectrophotometer.

## Statistical analysis

Data were analyzed using the program Stagraphics Centurion v. XVI (Statpoint Technologies, Warrenton, Virginia, USA). The mean and standard error (SE) were calculated from four replicates per cultivar (except for stem length, in which nine replicates were used). Mean values per species were calculated and the corresponding SE was calculated using cultivar means. The significance of the differences among control and drought stress treatments (expressed in percentage of variation over the control) for each variety was evaluated with Student's *t* tests. A multivariate principal components analysis (PCA) was performed using the cultivar means of both control and drought stress conditions in order to detect associations among the traits measured, as well as between the species and cultivars used. Data were standardized and Euclidean distances were used for the PCA.

## RESULTS

### Growth parameters

The stem length was longer in *T. tenuifolia* than in *T. patula* and *T. erecta* (Table 1) and decreased in the three species as a result of the water stress treatment, when compared with the non-stressed controls. On average, the smallest reduction was registered in *T. patula* (20.2%), and being pronounced in *T. tenuifolia* (27.8%) and in *T. erecta* (31.6%). Stem length reduction was statistically significant in all cultivars except in *T. erecta* 'Cupid Golden Yellow.' Large differences were observed among cultivars in the percentage of reduction of stem length (Table 1).

Fresh weight (FW) decreased significantly under drought stress in all cultivars of *T. patula* and *T. tenuifolia*, but only in one of *T. erecta* (Table 1). On average, a strong reduction of 78.8% was registered in *T. patula* and of 79.2% in *T. tenuifolia*, but of only 33.9% in *T. erecta*. In *T. patula*, the smallest reduction was registered in the cultivar 'Robuszta' (68.4%), whereas the largest in cv. 'Orion' (86.7%). Among cultivars of *T. tenuifolia*, the smallest loss in FW under water stress was registered in plants of 'Luna Gold' (49.0%), while the remaining three cultivars showed a dramatic reduction of FW of more than 90% with respect to their control FW. Among the *T. erecta* cultivars, the reduction in percentage ranged between 12.9% in 'Cupid Golden Yellow' and 43.9% in 'Alacsony Citromsarga.'

The water content (WC %) also decreased in drought-stressed plants when compared to their respective controls in all cultivars (Table 1). The water loss was about 18% in *T. patula* and *T. erecta* but reached 37.6% in *T. tenufolia*. Again, there were important differences within species. Among the cultivars of *T. patula*, the smallest reduction of only 6.6% was observed in cv. 'Bolero' and the highest of 27.4% in 'Orion.' In *T. tenuifolia*, the cultivar 'Luna Gold' registered a small reduction of 7.4% but plants of cv. 'Luna Orange' lost 68% of their water content under drought. A smaller variation of water loss was found in *T. erecta*, from 12.3% in 'Alacsony Citromsarga' to 26.5% in 'Cupid Golden Yellow.'

### Photosynthetic pigments

On average, total chlorophylls content was higher in *T. patula* than in the two other species, while the highest average level of carotenoids was observed in *T. erecta* (Table 2). However, important differences were found among cultivars of each single species. In the three analysed taxa, total chlorophylls and carotenoid concentrations decreased significantly, in all cultivars except *T. patula* cv. 'Bolero,' which maintained the same levels in stressed plants as those recorded in the control (Table 2). All other cultivars of *T. patula* showed a 50–70% reduction of photosynthetic pigments. In *T. tenuifolia*, 'Luna Gold' showed the smallest reduction and 'Luna Lemon' the highest for both types of pigments. In *T. erecta*, degradation of chlorophylls was similar in the three cultivars, but the decrease of total carotenoid contents was considerably smaller in 'Alacsony Citromsarga' than in the other two cultivars.

**Table 1** Stem length (SL), fresh weight (FW), and water content percentage (WC) values (mean ± SE) for control and drought-stressed (DS; three weeks without watering) plants of 12 cultivars of three species of *Tagetes*, and percentage of change of the drought stress treatment over the control.

| Cultivar | SL (cm) (n = 9) | | | FW (g) (n = 4) | | | WC (%) (n = 4) | | |
|---|---|---|---|---|---|---|---|---|---|
| | Control | DS | Change (%)[a] | Control | DS | Change (%)[a] | Control | DS | Change (%)[a] |
| *T. patula* | | | | | | | | | |
| 'Bolero' | 14.50 ± 0.89 | 11.83 ± 0.69 | −18.4* | 5.32 ± 0.30 | 1.68 ± 0.11 | −68.4*** | 91.91 ± 0.18 | 85.80 ± 0.48 | −6.6*** |
| 'Orange Flame' | 10.83 ± 0.47 | 9.00 ± 0.33 | −16.9** | 5.07 ± 0.51 | 1.38 ± 0.07 | −72.8*** | 92.49 ± 0.48 | 73.97 ± 0.50 | −20.0*** |
| 'Orion' | 9.11 ± 0.25 | 6.67 ± 0.12 | −26.8*** | 8.79 ± 0.51 | 1.17 ± 0.06 | −86.7*** | 91.25 ± 0.32 | 66.28 ± 2.96 | −27.4*** |
| 'Robuszta' | 8.17 ± 0.28 | 7.00 ± 0.32 | −14.3* | 5.19 ± 0.46 | 1.46 ± 0.24 | −71.9*** | 91.84 ± 0.21 | 76.00 ± 0.95 | −17.3*** |
| 'Szinkeverek' | 9.84 ± 0.17 | 7.33 ± 0.26 | −25.5*** | 8.68 ± 0.28 | 1.50 ± 0.12 | −82.7*** | 92.14 ± 0.26 | 73.82 ± 0.68 | −19.9*** |
| Mean | 10.49 ± 1.10 | 8.37 ± 0.96 | | 6.61 ± 0.87 | 1.44 ± 0.08 | | 91.93 ± 0.20 | 75.17 ± 3.14 | |
| *T. tenuifolia* | | | | | | | | | |
| 'Luna Gold' | 20.50 ± 1.21 | 16.94 ± 0.47 | −17.4* | 13.48 ± 0.08 | 6.87 ± 0.15 | −49.0*** | 91.02 ± 0.27 | 84.28 ± 1.80 | −7.4** |
| 'Luna Lemon' | 18.78 ± 0.50 | 13.17 ± 0.35 | −29.9*** | 6.07 ± 0.71 | 0.51 ± 0.03 | −91.6*** | 90.67 ± 0.12 | 39.12 ± 3.12 | −56.9*** |
| 'Luna Orange' | 18.83 ± 0.97 | 15.00 ± 0.38 | −20.3** | 7.89 ± 0.15 | 0.49 ± 0.01 | −93.8*** | 90.44 ± 0.44 | 28.96 ± 4.02 | −68.0*** |
| 'Sarga' | 24.72 ± 0.88 | 14.67 ± 0.58 | −40.7*** | 8.65 ± 0.14 | 0.61 ± 0.02 | −93.0*** | 91.33 ± 0.40 | 74.27 ± 1.59 | −18.7*** |
| Mean | 20.71 ± 1.40 | 14.95 ± 0.78 | | 9.02 ± 1.58 | 2.12 ± 1.58 | − | 90.87 ± 0.19 | 56.66 ± 13.38 | |
| *T. erecta* | | | | | | | | | |
| 'Alacsony Citromsarga' | 8.67 ± 0.68 | 5.44 ± 0.37 | −37.3*** | 4.12 ± 0.01 | 2.31 ± 0.09 | −43.9*** | 92.13 ± 0.74 | 80.84 ± 3.38 | −12.3* |
| 'Aranysarga' | 17.61 ± 0.53 | 11.44 ± 0.64 | −35.0*** | 7.30 ± 0.74 | 5.25 ± 0.42 | −28.1[n.s.] | 94.22 ± 0.69 | 77.78 ± 3.64 | −17.4** |
| 'Cupid Golden Yellow' | 6.50 ± 0.40 | 5.50 ± 0.29 | −15.4[n.s.] | 2.56 ± 0.21 | 2.23 ± 0.21 | −12.9[n.s.] | 91.54 ± 0.34 | 67.27 ± 2.53 | −26.5*** |
| Mean | 10.93 ± 3.40 | 7.46 ± 1.99 | | 4.66 ± 1.40 | 3.26 ± 0.99 | | 92.63 ± 0.81 | 75.3 ± 4.11 | |

**Notes:**
[a] According to Student t-test, the differences between control and drought plants of the same cultivar, are significant at;
*** $P < 0.001$
** $P < 0.01$
* $P < 0.05$
[n.s.] non-significant.

## Osmolyte accumulation

Proline (Pro) levels in control plants were highest in *T. tenuifolia*, intermediate in *T. patula* and lowest in *T. erecta* (Table 3). Water stress led to an increase in Pro concentrations in all cultivars, reaching 50–70-fold in *T. patula*, except for cv. 'Bolero,' which experienced a smaller increase, below 10-fold over the non-stressed control. In *T. tenuifolia*, a strong variation among cultivars, between 3 and 40-fold, was detected (Table 3). Also, in *T. erecta* cultivars, Pro levels in plants subjected to drought increased from approximately 40-fold in 'Alacsony Citromsarga' to almost 150-fold in 'Cupid Golden Yellow.'

For GB, the highest levels were observed in *T. tenuifolia*, both in control and stressed plants, and increased in the three species in response to water stress, from three to six-fold. However, the absolute contents of GB were much lower than those calculated for Pro in the same treated plants (Table 3). Total soluble sugar (TSS) concentrations were highest in *T. patula*, on average. Under drought conditions, an increase in TSS content of 1.5–2.5-fold in relation to the corresponding controls, was detected in *T. patula* and *T. tenuifolia* cultivars, and somewhat lower in the *T. erecta* plants (Table 3).

**Table 2** Total chlorophylls and total carotenoids values (mean ± SE) for control and drought-stressed (three weeks without watering) plants of 12 cultivars of three species of *Tagetes*, and percentage of change of the drought stress treatment over the control.

| Cultivar | Total chlorophylls (mg·g$^{-1}$ DW) (n = 4) | | | Total carotenoids (mg·g$^{-1}$ DW) (n = 4) | | |
|---|---|---|---|---|---|---|
| | Control | DS | Change (%)[a] | Control | DS | Change (%)[a] |
| *T. patula* | | | | | | |
| 'Bolero' | 11.44 ± 0.80 | 11.52 ± 0.75 | 0.7[n.s.] | 1.26 ± 0.11 | 1.19 ± 0.13 | −5.6[n.s.] |
| 'Orange Flame' | 13.87 ± 1.43 | 4.89 ± 0.15 | −64.8[***] | 1.64 ± 0.08 | 0.79 ± 0.06 | −52.0[***] |
| 'Orion' | 15.71 ± 0.70 | 3.44 ± 0.23 | −78.1[***] | 1.44 ± 0.07 | 0.41 ± 0.02 | −71.9[***] |
| 'Robuszta' | 15.35 ± 1.56 | 3.52 ± 0.44 | −77.1[***] | 1.51 ± 0.09 | 0.68 ± 0.06 | −54.9[***] |
| 'Szinkeverek' | 12.04 ± 0.87 | 4.48 ± 0.42 | −62.8[***] | 1.65 ± 0.11 | 0.51 ± 0.03 | −69.1[***] |
| Mean | 13.68 ± 0.86 | 5.57 ± 1.52 | | 1.50 ± 0.07 | 0.71 ± 0.14 | |
| *T. tenuifolia* | | | | | | |
| 'Luna Gold' | 9.69 ± 0.09 | 6.33 ± 0.43 | −34.6[***] | 1.39 ± 0.04 | 0.95 ± 0.05 | −31.7[***] |
| 'Luna Lemon' | 8.78 ± 0.48 | 1.65 ± 0.20 | −81.2[***] | 1.37 ± 0.07 | 0.25 ± 0.03 | −81.8[***] |
| 'Luna Orange' | 11.89 ± 1.46 | 2.98 ± 0.12 | −74.9[***] | 1.50 ± 0.07 | 0.39 ± 0.03 | −73.7[***] |
| 'Sarga' | 11.57 ± 0.89 | 5.51 ± 0.38 | −52.4[***] | 1.60 ± 0.07 | 0.80 ± 0.08 | −49.7[***] |
| Mean | 10.48 ± 0.75 | 4.12 ± 1.09 | | 1.46 ± 0.05 | 0.60 ± 0.17 | |
| *T. erecta* | | | | | | |
| 'Alacsony Citromsarga' | 13.13 ± 0.62 | 4.45 ± 0.56 | −66.1[***] | 1.99 ± 0.12 | 1.40 ± 0.11 | −29.9[*] |
| 'Aranysarga' | 9.24 ± 0.10 | 3.22 ± 0.48 | −65.1[***] | 2.28 ± 0.13 | 0.95 ± 0.08 | −58.2[***] |
| 'Cupid Golden Yellow' | 9.26 ± 0.57 | 4.02 ± 0.42 | −56.6[***] | 2.11 ± 0.18 | 0.79 ± 0.08 | −62.5[***] |
| Mean | 10.54 ± 1.29 | 3.90 ± 0.36 | | 2.13 ± 0.08 | 1.04 ± 0.18 | |

**Notes:**
[a] According to Student t-test, the differences between control and drought plants of the same cultivar, are significant at;
[***] $P < 0.001$
[**] $P < 0.01$
[*] $P < 0.05$
[n.s.] non-significant.

## Malondialdehyde and non-enzymatic antioxidants

Average MDA levels were higher in *T. tenuifolia* and *T. erecta* than in *T. patula* and increased slightly in the stressed plants of most *Tagetes* cultivars under study (Table 4). However, for one cultivar of *T. patula* ('Bolero') and all cultivars of *T. tenuifolia* except 'Sarga,' the differences were not significant (Table 4). Total phenolic compounds (TPC) and total flavonoids (TF) were higher in *T. patula* and *T. tenuifolia* than in *T. erecta* and increased in stressed plants of the three species, but again, this increment was generally low (1.06–1.65-fold for TPC and 1.02–1.51-fold for TF), and it was not significant in several cultivars under study (Table 4).

## Multivariate analysis

The first and second components of the PCA accounted for 48.7 and 20.9%, respectively, of the total variation observed. The first component displayed a strong positive correlation with plant growth traits: fresh weight and water content, as well as with

**Table 3** Proline, fresh weight (FW), glycine betaine (GB) and total soluble sugars (TS) values (mean ± SE) for control and drought-stressed (DS; three weeks without watering) plants of 12 cultivars of three species of *Tagetes*, and percentage of change of the drought stress treatment over the control.

| Cultivar | Proline ($\mu$mol·g$^{-1}$ DW) (n = 4) | | | GB ($\mu$mol·g$^{-1}$ DW) (n = 4) | | | TSS (mg eq. glucose g$^{-1}$ DW) (n = 4) | | |
|---|---|---|---|---|---|---|---|---|---|
| | Control | DS | Change (%) | Control | DS | Change (%) | Control | DS | Change (%) |
| *T. patula* | | | | | | | | | |
| 'Bolero' | 6.86 ± 1.03 | 67.34 ± 8.41 | 881.7*** | 2.15 ± 0.22 | 9.10 ± 0.15 | 323.9*** | 21.13 ± 1.62 | 49.53 ± 5.49 | 134.4** |
| 'Orange Flame' | 5.31 ± 0.99 | 250.35 ± 24.35 | 4,610.8*** | 2.40 ± 0.07 | 11.00 ± 0.76 | 358.3*** | 24.83 ± 1.90 | 36.51 ± 3.70 | 47.0* |
| 'Orion' | 3.88 ± 0.88 | 261.49 ± 28.72 | 6,635.0*** | 3.08 ± 0.22 | 11.40 ± 0.89 | 269.8*** | 24.52 ± 3.80 | 36.00 ± 1.62 | 46.8* |
| 'Robuszta' | 4.48 ± 0.87 | 275.76 ± 22.68 | 6,053.2*** | 2.33 ± 0.17 | 6.09 ± 0.31 | 161.4*** | 27.52 ± 4.29 | 35.79 ± 2.47 | 30.1[n.s.] |
| 'Szinkeverek' | 6.17 ± 0.70 | 363.32 ± 11.59 | 5,784.7*** | 1.51 ± 0.18 | 7.99 ± 0.52 | 427.9*** | 15.64 ± 1.82 | 29.36 ± 1.68 | 87.7** |
| Mean | 5.34 ± 0.54 | 243.65 ± 48.51 | | 2.29 ± 0.25 | 9.11 ± 0.98 | | 22.73 ± 2.05 | 37.44 ± 3.30 | |
| *T. tenuifolia* | | | | | | | | | |
| 'Luna Gold' | 15.61 ± 1.55 | 45.14 ± 0.56 | 189.2*** | 7.07 ± 0.32 | 12.83 ± 0.95 | 81.4 | 9.79 ± 1.16 | 24.50 ± 2.80 | 150.3** |
| 'Luna Lemon' | 7.23 ± 0.76 | 230.75 ± 23.99 | 3,090.3*** | 3.21 ± 0.21 | 19.56 ± 2.07 | 508.9*** | 16.54 ± 1.68 | 24.44 ± 2.43 | 47.8* |
| 'Luna Orange' | 6.21 ± 0.08 | 244.63 ± 19.06 | 3,836.6*** | 4.92 ± 0.57 | 15.23 ± 1.18 | 209.6*** | 17.96 ± 0.71 | 29.89 ± 3.47 | 66.5* |
| 'Sarga' | 8.37 ± 0.50 | 88.08 ± 3.92 | 952.9*** | 3.42 ± 0.14 | 14.81 ± 0.45 | 332.5*** | 18.23 ± 1.51 | 27.76 ± 0.99 | 52.3** |
| Mean | 9.36 ± 2.13 | 152.15 ± 50.24 | | 4.66 ± 0.89 | 15.60 ± 1.42 | | 15.63 ± 1.98 | 26.65 ± 1.33 | |
| *T. erecta* | | | | | | | | | |
| 'Alacsony Citromsarga' | 2.48 ± 0.18 | 97.05 ± 3.65 | 3,819.2*** | 3.28 ± 0.14 | 15.86 ± 0.84 | 383.9*** | 13.69 ± 0.90 | 16.83 ± 1.29 | 23.0[n.s.] |
| 'Aranysarga' | 2.77 ± 0.24 | 248.06 ± 3.46 | 8,867.7*** | 2.80 ± 0.32 | 7.40 ± 0.42 | 163.8*** | 19.82 ± 0.90 | 22.65 ± 1.31 | 14.3[n.s.] |
| 'Cupid Golden Yellow' | 2.30 ± 0.19 | 343.45 ± 20.32 | 14,810.8*** | 3.57 ± 0.35 | 11.40 ± 0.67 | 219.4*** | 11.73 ± 0.91 | 16.02 ± 1.95 | 36.5[n.s.] |
| Mean | 2.52 ± 0.14 | 229.52 ± 71.82 | | 3.22 ± 0.22 | 11.55 ± 2.45 | | 15.08 ± 2.44 | 18.50 ± 2.09 | |

**Notes:**
[a] According to Student t-test, the differences between control and drought plants of the same cultivar, are significant at;
*** $P < 0.001$
** $P < 0.01$
* $P < 0.05$
[n.s.] non-significant.

photosynthetic pigments, while it showed a negative correlation with osmolytes and the analyzed antioxidants (Fig. 1). The second principal component was highly correlated with total chlorophylls, total soluble sugars and antioxidants, and negatively correlated with the oxidative stress marker MDA.

The PCA plot with the mean values of cultivars tested under control and drought stress conditions clearly separates the control materials, which have positive values for this first component, from the materials subjected to drought stress (Fig. 2). The cultivars that under the drought stress treatment had higher values (i.e., closer to non-stressed controls) were 'Bolero' for *T. patula*, 'Luna Gold' for *T. tenuifolia* and the three cultivars of *T. erecta*. Furthermore, within *T. patula* and *T. tenuifolia*, a large dispersion is observed among the cultivars subjected to stress in this first component (Fig. 2). The second component separates the three species, with *T. patula* displaying higher values, *T. tenuifolia* intermediate ones and *T. erecta* the lowest values.

**Table 4** Malondialdehyde (MDA), total phenolic compounds (TPC), and total flavonoids (TF) values (mean ± SE) for control and drought-stressed (DS; three weeks without watering) plants of 12 cultivars of three species of *Tagetes*, and percentage of change of the drought stress treatment over the control.

| Cultivar | MDA ($nmol·g^{-1}$ DW) (n = 4) | | | TPC (mg eq. $GA·g^{-1}$ DW) (n = 4) | | | TF (mg eq. $C·g^{-1}$ DW) (n = 4) | | |
|---|---|---|---|---|---|---|---|---|---|
| | Control | DS | Change (%) | Control | DS | Change (%) | Control | DS | Change (%) |
| *T. patula* | | | | | | | | | |
| 'Bolero' | 282.43 ± 46.09 | 279.47 ± 23.86 | −1.0[n.s.] | 5.47 ± 0.92 | 6.53 ± 0.93 | 19.4[n.s.] | 6.84 ± 0.34 | 7.13 ± 0.74 | 4.2[n.s.] |
| 'Orange Flame' | 312.10 ± 4.91 | 350.91 ± 53.06 | 12.4[n.s.] | 7.16 ± 1.03 | 8.57 ± 0.79 | 19.7[n.s.] | 6.83 ± 0.14 | 8.34 ± 0.33 | 22.1[**] |
| 'Orion' | 254.22 ± 26.93 | 351.07 ± 4.51 | 38.1[*] | 7.25 ± 1.18 | 8.46 ± 0.82 | 16.6[n.s.] | 7.21 ± 0.62 | 9.29 ± 0.43 | 28.8[*] |
| 'Robuszta' | 216.18 ± 7.50 | 307.06 ± 9.34 | 42.0[***] | 4.63 ± 0.23 | 7.68 ± 0.27 | 65.8[***] | 6.74 ± 0.35 | 9.05 ± 0.34 | 34.2[*] |
| 'Szinkeverek' | 365.39 ± 36.77 | 388.06 ± 29.07 | 6.2[n.s.] | 6.27 ± 0.12 | 7.20 ± 0.50 | 14.9[n.s.] | 7.88 ± 0.28 | 8.66 ± 0.50 | 9.9[n.s.] |
| Mean | 286.06 ± 25.45 | 335.31 ± 19.01 | | 6.16 ± 0.50 | 7.69 ± 0.38 | | 7.10 ± 0.21 | 8.49 ± 0.38 | |
| *T. tenuifolia* | | | | | | | | | |
| 'Luna Gold' | 431.65 ± 5.09 | 437.98 ± 29.87 | 1.5[n.s.] | 7.13 ± 0.82 | 8.22 ± 0.13 | 15.2[n.s.] | 4.68 ± 0.42 | 4.79 ± 0.60 | 2.3[n.s.] |
| 'Luna Lemon' | 428.71 ± 30.84 | 431.50 ± 16.13 | 0.6[n.s.] | 6.51 ± 1.15 | 7.06 ± 0.33 | 8.5[n.s.] | 5.20 ± 0.59 | 7.85 ± 0.59 | 51.0[*] |
| 'Luna Orange' | 432.29 ± 26.65 | 451.17 ± 18.37 | 4.4[n.s.] | 8.38 ± 0.35 | 10.03 ± 0.79 | 19.7[n.s.] | 5.52 ± 0.38 | 7.70 ± 0.80 | 39.5[*] |
| 'Sarga' | 512.34 ± 39.56 | 639.19 ± 16.53 | 24.8[*] | 6.87 ± 0.51 | 7.34 ± 0.23 | 6.8[n.s.] | 4.83 ± 0.23 | 7.31 ± 0.92 | 51.5[*] |
| Mean | 451.25 ± 20.38 | 489.96 ± 49.91 | | 7.22 ± 0.41 | 8.16 ± 0.67 | | 5.06 ± 0.19 | 6.91 ± 0.72 | |
| *T. erecta* | | | | | | | | | |
| 'Alacsony Citromsarga' | 414.50 ± 50.48 | 440.35 ± 22.40 | 6.2[n.s.] | 4.13 ± 0.37 | 4.40 ± 0.39 | 6.7[n.s.] | 2.99 ± 0.26 | 3.21 ± 0.14 | 7.6[n.s.] |
| 'Aranysarga' | 451.88 ± 26.47 | 559.07 ± 17.20 | 23.7[*] | 3.91 ± 0.33 | 4.13 ± 0.09 | 5.5[n.s.] | 2.72 ± 0.19 | 3.27 ± 0.09 | 20.3[*] |
| 'Cupid Golden Yellow' | 408.89 ± 38.79 | 574.18 ± 58.67 | 40.4[n.s.] | 2.52 ± 0.25 | 3.14 ± 0.31 | 24.6[n.s.] | 2.83 ± 0.31 | 3.00 ± 0.16 | 6.0[n.s.] |
| Mean | 425.09 ± 13.51 | 524.53 ± 42.37 | | 3.52 ± 0.50 | 3.90 ± 0.38 | | 2.85 ± 0.08 | 3.16 ± 0.08 | |

**Notes:**

[a] According to Student t-test, the differences between control and drought plants of the same cultivar, are significant at;

[***] $P < 0.001$

[**] $P < 0.01$

[*] $P < 0.05$

[n.s.] non-significant.

## DISCUSSION

Plant water relations are strongly affected by water deficit, as drought has a direct effect on photosynthesis and growth rate (*Chaves & Oliveira, 2004*). Severe drought reduces the rate of cell division and cell enlargement, affects enzyme activities, causes loss of turgor and a drop in the synthesis of carbohydrates through photosynthesis (*Farooq et al., 2012* and references within). Reduction of growth under drought stress conditions is more pronounced in stress sensitive plants than in the stress tolerant ones (*Demiral & Türkan, 2005*) and, in consequence, the selection of cultivars tolerant to drought is needed when drought stress is expected. This has a particular importance in the Mediterranean region, characterized by harsh climatic conditions in summer, which probably will worsen in the near future due to the global warming (*Rubio, 2009*).

Among herbaceous ornamentals, the species of the genus *Tagetes* are considered as relatively drought resistant, as they are native to areas with hot and dry climate (*Henson, Newman & Hartley, 2006*). However, the results obtained here indicate that there is a large

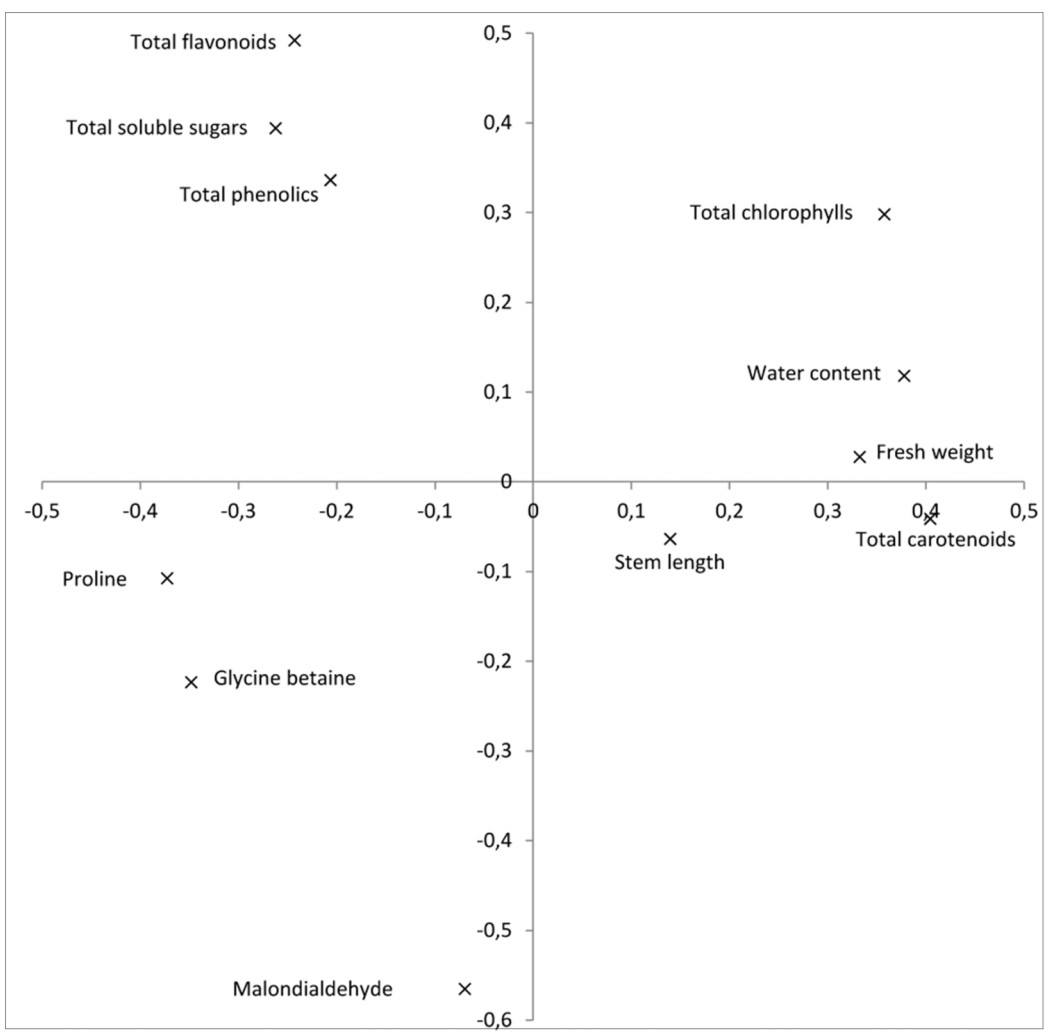

**Figure 1** Relationships between the plant growth characteristics (stem length, fresh weight, and water content), photosynthetic pigments (total chlorophylls, total carotenoids), soluble solutes (proline, glycine betaine, and total soluble solutes), oxidative stress marker MDA and anti-oxidants (total phenolics and total flavonoids) based on the two first components of the PCA (accounting for 48.7 and 20.9% of the total variation, respectively).

variation among marigold species regarding their ability to respond to severe water stress. According to different evaluated parameters, related to tolerance to drought, the most tolerant of the three species proved to be *T. erecta*. When considering cultivars within species, *T. patula* and *T. tenuifolia*, lost more than double of their FW under stress, in comparison to *T. erecta*. This confirms previous findings that reported this species as drought tolerant (*Henson, Newman & Hartley, 2006*; *Riaz et al., 2013*), and even as able to withstand moderate salinity (*Valdez-Aguilar, Grieve & Poss, 2009*; *Sayyed et al., 2014*). The most affected by stress were cultivars of signet marigold (*T. tenuifolia*), and *T. patula* was in an intermediate position, although both species have been considered as tolerant to short-time drought (*Borch et al., 2003*; *Kafi & Jouyban, 2013*). When comparing the stem length, signet marigold cultivars had considerably longer stems than the others in

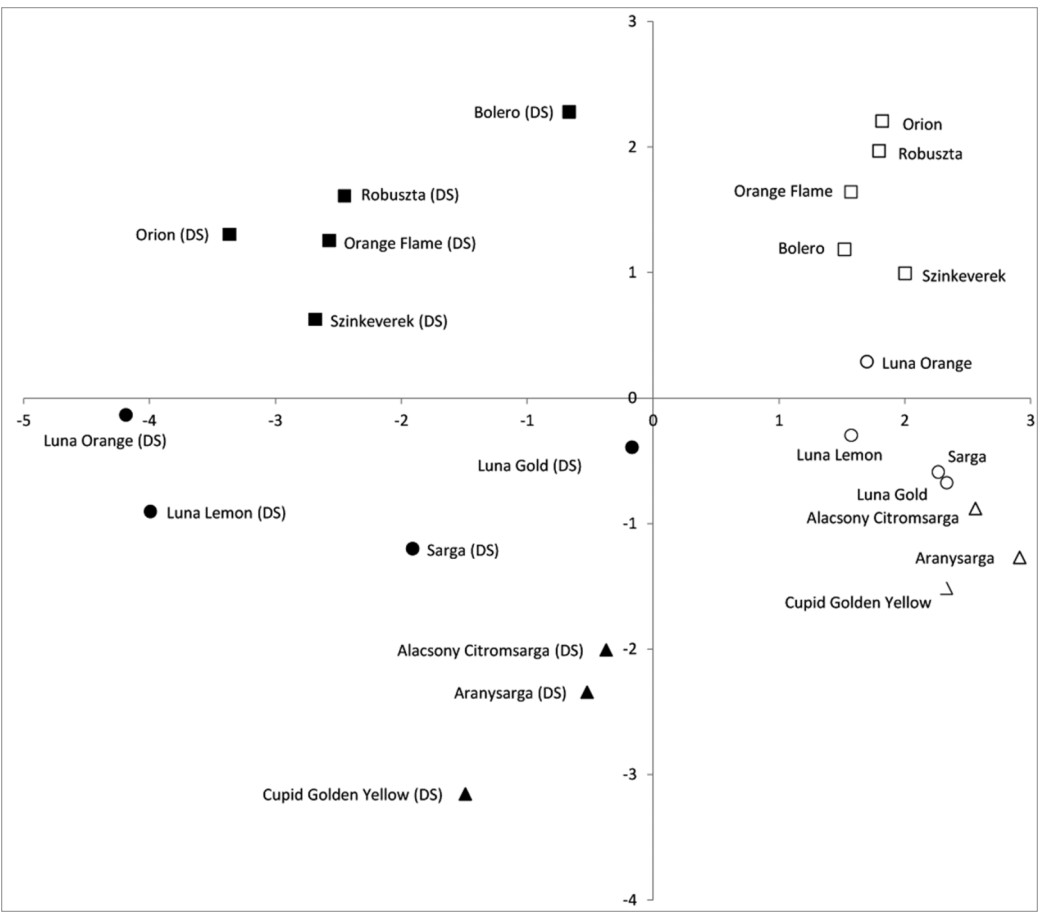

**Figure 2 Similarities based on plant growth and biochemical marker traits among controls and drought-stressed materials of 12 varieties of three *Tagetes* species (*T. patula*, *T. tenuifolia* and *T. erecta*) based on the two first components of the PCA (accounting for 48.7 and 20.9% of the total variation, respectively).** Control and drought-stressed materials are represented by white and black symbols, respectively. Additionally, "(DS)" is indicated in the drought-stressed materials. Cultivars of *T. patula* are represented by squares, those of *T. tenuifolia* by circles and those of *T. erecta* by triangles).

both control and drought stress treatments. The rate of growth, implicitly the size of plants, is directly related to the effect of water stress.

Besides interspecific differences, drought tolerance largely varies among genotypes within crop species (*Gholami, Rahemi & Rastegar, 2012*; *Cortés et al., 2012*; *Siddiqui et al., 2015*; *Yang et al., 2015*). Also, it is known that seedlings or young plants are generally more affected by stress than plants at later developmental stages (*Vicente et al., 2004*). Therefore, selection in early stages may be an efficient way to select drought tolerant cultivars of *Tagetes*.

Cultivars 'Szinkeverek,' 'Robuszta,' and 'Bolero' of *T. patula* showed a smaller stress-induced reduction of SL and FW, but the latter also appeared to be more resistant to dehydration, since water loss under drought conditions was relatively lower. Therefore, we assume that cv. 'Bolero' is the most appropriate for cultivation in areas or conditions where drought may occur frequently. In the species *T. tenuifolia*, the genotype 'Luna Gold'

was clearly the most drought tolerant of the four cultivars analyzed here. Among the cultivars of *T. erecta*, the smallest reduction in LS and FW was registered in 'Cupid Golden Yellow,' although it showed a more pronounced reduction in its WC than the other cultivars.

Although growth parameters are reliable and commonly used to assess the effects of stress in most plant species, they can be complemented or even substituted by suitable biochemical markers, which include a large array of compounds that can be easily and quickly identified (*Schiop et al., 2015*). In addition, the measurement of these biochemical markers can be made without the destruction of the plant required for measuring some growth traits, such as fresh weight. Abiotic stresses, including drought, lead to the degradation of photosynthetic pigments (*Parida & Das, 2005*). Monitoring the decrease in the contents of photosynthetic pigments in affected plants can be used as a biomarker of stress (*Schiop et al., 2015*), There are numerous reports of decreased levels of chlorophylls and carotenoids under water stress in different species (i.e., *Logini et al., 1999*; *Agastian, Kingsley & Vivekanandan, 2000*; *Al Hassan et al., 2015*; *Yang et al., 2015*), including *T. erecta* (*Riaz et al., 2013*), thus our findings confirm that in the *Tagetes* species evaluated, the reduction in the levels of photosynthetic pigments matches the inhibition of growth in the different cultivars.

Osmotic stress induces cellular accumulation of compatible solutes, or osmolytes in all plants, regardless of them being tolerant or not to abiotic stress (*Ashraf & Foolad, 2007*). One ubiquitous osmolyte in plants is proline, which is synthesised under many different environmental stress conditions, such as salinity, drought, cold, high temperature, nutritional deficiencies, heavy metals, air pollution or high UV radiation (*Hare, Cress & van Standen, 1998*; *Grigore, Boscaiu & Vicente, 2011*). In addition to its role in osmotic adjustment, Pro plays many other functions, as 'osmoprotectant,' directly stabilising proteins, membranes and other subcellular structures, scavenging free radicals, or balancing the cell redox status under stress conditions (*Smirnoff & Cumbes, 1989*; *Verbruggen & Hermans, 2008*), as well as contributing to storage of carbon and nitrogen during stress, which will help cell and tissue recovery when stress eases or disappears (*Szabados & Savouré, 2010*). Studies on congeneric wild species or on different cultivars of the same species highlight that although higher Pro levels can be found in the more tolerant taxa (*Boscaiu et al., 2013*; *Ghanbari et al., 2013*), there is often no positive correlation between Pro contents and the relative degree of tolerance (*Ashraf & Foolad, 2007*; *Chen et al., 2007*). In all cultivars analysed here, there was a significant increase in Pro, some reaching very high levels, as reported previously in other *Tagetes* cultivars (*Mohamed, Harris & Henderson, 2000*). The concentrations that we measured indicate that Pro plays an important role in osmotic adjustment in *Tagetes*. However, there is only a slight correlation of Pro levels and the reduction of the water content of plants under drought, and not statistically significant. In the French marigold, WC did not change significantly in cv. 'Bolero' and a much smaller increase was registered in Pro levels in comparison with the other cultivars of this species. In signet marigold, the two cultivars that had a higher water loss ('Luna Orange' and 'Luna Lemon') accumulated much higher amounts of Pro, and in the African marigold, the highest level of Pro was registered in 'Cupid Golden Yellow,' which reduced more its WC under drought than the two other cultivars.

GB is a quaternary ammonium compound synthesised in response to salt and water stress in many different plant groups (*Hanson & Scott, 1980*; *Rhodes & Hanson, 1993*; *Ashraf & Foolad, 2007*). There are no previous reports on the levels and role of this osmolyte in marigold, but we found that it increases significantly under drought in all genotypes. However, the values that we detected are by far much lower than those found in plants that are true GB accumulators (*Khan, Ungar & Showalter, 2000*; *Gil et al., 2013*). Carbohydrates also play an important role as osmolytes in many plants species (*Gil et al., 2011*); in *Tagetes*, total soluble sugars contents increased in response to drought, in all tested cultivars.

MDA is a product of membrane lipid peroxidation, considered a reliable general marker of oxidative stress (*Del Rio, Stewart & Pellegrini, 2005*), and it is routinely used to assess the degree of oxidative damage induced in plants by different stress treatments. In comparative analyses, more tolerant cultivars generally exhibit a smaller amount of MDA, as reported in many species (e.g., *Hussain et al., 2013*). In the present study, MDA increase was not significant in all cultivars and its levels could not be correlated with the degree of tolerance. On the other hand, phenolic compounds and, especially, a subgroup of them, the flavonoids, are plant secondary metabolites which are important in the mechanisms of adaptation to abiotic stresses (*Di Ferdinando et al., 2012*; *Di Ferdinando et al., 2014*), among many other biological functions. Since many flavonoids and other phenolic compounds are strong antioxidants, their accumulation in plants can reduce oxidative damage induced by different abiotic stresses, including drought (*Bautista et al., 2016*). Flavonoids are regarded as a secondary ROS scavenging system activated in plants under severe stress because of the depletion of primary antioxidant defence systems. Therefore, the biosynthesis of antioxidant flavonoids is triggered, especially under severe stress conditions, when the activities of antioxidant enzymes, considered the first line of defence against ROS, decline (*Fini et al., 2011*). Cultivars of the species *T. erecta* are known for their high phenolic and flavonoid contents and radical-scavenging activity (*Li et al., 2007*). Water stress induced a significant increase in the levels of total phenolic compounds and total flavonoids in all genotypes. Total antioxidant flavonoids generally correlated better than total phenolics with the morphological markers of stress.

The combined analysis through a PCA of the different parameters evaluated allowed a clear identification of the most tolerant cultivars in each species, which are those that plot closer to the non-stressed materials, as well as those traits that present a greater association with drought tolerance. This approach, based on traits measured in young plants, may allow the early identification of drought stress tolerant cultivars of *Tagetes*.

## CONCLUSIONS

We have identified several tolerant cultivars to drought in three ornamental *Tagetes* species. The large variation observed among species and among cultivars within species indicates that there are good prospects for the selection of cultivars with enhanced tolerance to drought. The degradation of photosynthetic pigments and changes in the levels of osmolytes, total phenolics and total flavonoids are suitable markers for testing the tolerance to drought in *Tagetes* cultivars. Among osmolytes, the most reliable stress marker

is proline: higher levels of Pro were detected in the more sensitive genotypes of the same species—and, therefore, those more affected by drought. As a response to oxidative stress, an increase in total phenolic compounds and total flavonoids was detected, generally lower in the more tolerant cultivars. Our study revealed that, when comparing different species belonging to the same genus, it is more appropriate the simultaneous use of several biochemical markers, as this allows the identification of tolerant cultivars. Since the amount of plant material required for these biochemical analyses is very low, they can be used in early seedling stages, without the necessity of growing the plants and analysing their growth response, which can be more laborious and time consuming. Our results contribute to the methods of screening for drought tolerance in *Tagetes*, an important objective for the production of plants of this ornamental species with improved water efficiency and adapted to a climate change scenario, especially in the Mediterranean region.

## ABBREVIATIONS

| | |
|---|---|
| **Car** | carotenoids |
| **Chl** | a + b total chlorophylls |
| **FW** | fresh weight |
| **GB** | glycine betaine |
| **MDA** | malondialdehyde |
| **Pro** | proline |
| **SL** | stem length |
| **TF** | total flavonoids |
| **TPC** | total phenolic compounds |
| **TSS** | total soluble sugars |
| **WC** | water content |
| **WS** | water stress |

### Funding
The authors received no funding for this work.

### Competing Interests
The authors declare that they have no competing interests.

### Author Contributions
- Raluca Cicevan performed the experiments, analyzed the data.
- Mohamad Al Hassan performed the experiments, analyzed the data, wrote the paper, prepared figures and/or tables.
- Adriana F. Sestras analyzed the data.
- Jaime Prohens analyzed the data, wrote the paper, prepared figures and/or tables, reviewed drafts of the paper.

- Oscar Vicente conceived and designed the experiments, contributed reagents/materials/analysis tools, wrote the paper, reviewed drafts of the paper.
- Radu E. Sestras conceived and designed the experiments, reviewed drafts of the paper.
- Monica Boscaiu conceived and designed the experiments, wrote the paper, prepared figures and/or tables, reviewed drafts of the paper.

## Data Deposition

The raw data has been supplied as Supplemental Dataset Files.

## Supplemental Information

Supplemental information for this article can be found online at http://dx.doi.org/10.7717/peerj.2133#supplemental-information.

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
