# Peer review of "Screening for drought tolerance in cultivars of the ornamental genus Tagetes (Asteraceae)"

_PeerJ, doi:10.7717/peerj.2133_

## Round 0.1 · original submission · Minor Revisions

There are only a few but pertinent comments on the text, that should be addressed in the reviewed version.

Reviewer 1 ·

Basic reporting

The article written in good and standard English. Manuscript include sufficient introduction and background to demonstrate the present study. Figures are relevant to the content of the article.

Experimental design

The present study described original primary research within the scope of of the journal. Research question is clearly defined which is relevant and meaningful in the field of stress physiology. Established protocols applied on new genus Tagetus. Authors followed prevailing ethical standards in the field.

Validity of the findings

The discussion part must be improved in the light of recent literature. The concluisons are appropriately stated.

Additional comments

No comments

·

Basic reporting

The manuscript represents a useful study using cheap and simple methodological approaches (spectrophotometrric determination of total chlorophylls and carotenoids, proline, etc.) to assess drought response in 12 Tagetes cultivars. The mastery lies in multivariate statsitical analyses of all the obtained data applying principal component analysis approach.

Experimental design

Simple, cheap, but efficient methods are used to determine basic biochemical characteristics in drought-treated plants such as total chlorophylls and carotenoids, total soluble sugars, total flavonoids, phenolics, proline, glycienbetaine, and malondialdehyde content.

I ahve only one note on the manuscript:

The spectrophotometer type and the manufacturer (company) used for spectrometric analyses has to be specified.

Validity of the findings

The data presented have appropriate statistical evaluation. Not only the individual parameters determined were evaluated using Student T-test (control versus drought-treated plants), but also the whole sample set was subjecetd to multivariate principal component analysis to evaluate the similar as well as the different trends between the individual growth and biochemical characteristics determined (stem length, fresh weight, total chlorophylls, total carotenoids, water content, glycine betaine, proline, malondialdehyde, total flavonoids, total phenolics, total soluble sugars) as well as between the individual marigold genoytpes.

Additional comments

Reviewer comments PeerJ-9597

The manuscript entitled „Screening for drought tolerance in cultivars of the ornamental genus Tagetes (Asteraceae)“ represents a valuable study on key plant growth characteristics and biochemical components (the levels of total chlorophylls and carotenoids, accumulation of soluble solutes proline, glycine betaine and total sugars, determination of total flavonoids, total phenolics and malondialdehyde content) using simple determination methods and measurements. I appreciate the use of cheap, simple methods and the complex statisdtical evaluation of the obtained results using multivariate principal component analysis. The application of a multivariate statistical analysis (principal component analysis) in data analysis enabled to show which physiologicl and biochemical characteristics reveal similar trends in the given sample sets as well as which marigold genotypes behaved similarly under the given environmental conditions.
I have no major comments on the manuscript.
I have only a few formal comments on the manuscript, mostly on the references.
There are several discrepancies between the references cited in the manuscript text and the references listed in the Reference list.
Line 73: Correct the spelling of the author´s name „Raghuveer et al. 2011“ according to the Reference list (not „Raghuuver et al., 2011“ as cited on line 73).
Line 403: „Al Hassan et al. 2014“ versus „Al Hassan et al. 2015“ in the Reference list.
Line 104: „Gholinezhad et al. 2014“ versus „Gholiezhad et al. 2014“ in the Reference list.
Line 411: „Hare and Cress 1997“ versus „Hare et al. 1998“ in the Reference list.
Line 124: „Jimenez et al. 2013“ - a reference which is missing in the Reference list.
Lines 88, 91: „Serrato-Cruz 2005“ versus „Serrato-Cruz 2004“ in the Reference list.
Extra references in the Reference list which are not cited in the manuscript text and thus have to eb removed: „Li et al. 2006“, „Santos et al. 2009“.

Further formal comments:
Introduction, lines 83, 84: Use the word „of“ instead of „with“ and insert the word „than“ in the sentence „…which have been developer give plants of a small size, usually not more than 40 cm tall,…“
Introduction, line 108: Insert a comma in the sentence „In addition, on many occasions,…“
Line 109: Insert „a“ before the words „lack of regular watering by custemrs“, i.e., ůa lack of watering by customers.“
Line 113: Insert a comma after the word „Nowadays,…“
In Materials and methods, spectrophotometer type including manufacturer (company) specification used for sprectrometric analyses of proline, total chlorophyllsand carotenoids has to be given.
Line 169: In „chlorophyll a, chlorophyll b“, „a“ and „b“ have to be written in italics.
Results, line 251: Insert a comma following the word „Again,…“
Line 252: Insert a comma following the words „Among the cultivars of T. patula,…“
In the table legends (Table 1, 2, 3, 4), statistical test used for the data evaluation, i.e., „Student T-test“ has to be specified regarding the significant as well as non-significant differences between control and drought-treated samples.
Discussion,
Line 362: Insert a comma before and after the words „in consequence“…“and, in consequence, the selection of cultivars…“
Line 375: Modify the word „stand“ to „withstand“ in the sentence „…and even able to withstand moderate salinity…“
Line 399: Use the verb „lead to“ instead of „produce“ in the sentence „Abiotic stresses, including drought, lead to the degradation of photosynthetic pigments…“
Line 405: Insert a comma following the words „…thus our findings confirm that in the Tagetes species evaluated,…“
Line 407: Remove „the“ in the sentence „Osmotic stress induces cellular accumulation of compatible solutes,…“
Line 417: Correct the word form „congener“ to „congeneric“ in the sentence „Studie son congeneric wild species…“
Line 429: Insert a comma before and after the words „in the African marigold,…“ in the sentence „…and, in the African marigold, the highest elvel of Pro was registered…“
Line 452: Insert commas in the sentence „Therefore, the biosynthesis of antioxidant flavonoids is triggered, especially under severe stress conditions,…“

Final recommendation: Accept after a minor (formal) revision.

Reviewer 3 ·

Basic reporting

The manuscript reports on the assessment of drought tolerance in cultivars of three species of Tagetes: i.e. T. patula, T. tenuifolia and T. erecta and on the identification biochemical markers for its quick assessment.
The effect of drought stress was evaluated by measuring growth parameters (stem length, fresh and dry weight of the leaves and water content) as well biochemical indicators, i.e.: photosynthetic pigments (chlorophyll a, chlorophyll b and total carotenoids), osmolytes (proline, glycine betaine, total soluble sugars), oxidative stress marker (malondialdehyde, a final product of membrane lipid peroxidation) and total antioxidant flavonoids and phenolic compounds.
The manuscript adhere to all PerrJ policies, it is well written and reports original results. The title clearly reflect the findings of the manuscript. The introduction appropriately reports the state of the art and the objectives are clearly defined. The results are of practical interest and critically discussed with reference to relevant literature . Figures and tables are clear, relevant and of good quality.

Experimental design

Methods are clearly described, with sufficient information to be reproducible by other investigators. Statistical analyses of data are properly performed .

Validity of the findings

The results reported in the manuscript are of practical interest. Considerable differences in water stress tolerance among and within Tagete species were detected, and the most tolerant cultivars in each of the species in study were identified. Furthermore, useful biochemical markers for an early screening of genotypes tolerant to drought were recognized
Drought stress was found to cause a marked reduction in plant growth and content in carotenoid pigments in the cultivars in study, as well as an increase in oxidative stress and soluble solutes. In particular, an increase in proline content resulted the most reliable water stress marker among the osmolytes tested.
The conclusions are well stated and connected to the original question investigated.

Additional comments

No Comments

·

Basic reporting

No comments

Experimental design

No comments

Validity of the findings

No comments

Additional comments

This manuscript reports the correlations between drought tolerance and different physiological traits and metabolites in 12 cultivars from three species of the genus Tagetes (Asteraceae), identifying the more drought tolerant cultivars and the more promising physiological markers for selection. The research is rigorous, and data are robust and statistically sound. The research topic is of great relevance in order to identify early markers to develop screening methodologies for the selection of drought tolerance genotypes, which is increasingly interesting to address global climate change in this and other species. The manuscript is very well written, following the specifications of PeerJ, and the Figures and Tables are relevant and properly labelled. So, I encourage the publication of this manuscript in PeerJ. It requires few changes before accepting it for publication in this Journal.
Minor changes:
- Tables. Please indicate that *, **, etc. indicate significant differences between control and drought stressed plants of the same cultivar at P=……..
- In some parts of the text Figures were cited as Fig. X, and in other as Figure X. Please conform to Journal standard.

---

## Round 0.2 · accepted · Accept

There are still some typos mistakes as you will see in the reviewers' comments. Please take that into account when doing the proof check

·

Basic reporting

The manuscript provides all relevant literature information and experimental data regarding Tagetes response to drought determined by growth and biochemical characteristics.

Experimental design

The experimental design is described adequately including all necessary information.

Validity of the findings

All results and conclusions are underlined by relevant experimental data.

Additional comments

Reviewer comments PeerReview Journal
The revised manuscript „Screening for drought tolerance in cultivars of the ornamental genus Tagetes (Asteraceae)“ is suitable for publication.
I have only a few minor formal comments on the manuscript.
Final decision: Accept afetr a minor (formal) revision.

Minor formal comments:
Line 97: Insert a comma following the words „Climatic scenarios predict that by the end of this century,…“
Line 102: Insert a comma following the words „Considering that in the near future,…“
Line 124: Insert a comma following the words „but under environmental stress conditions,…“
Lines 181, 185, 189, 208: Remove „the“ before the word „absorbance“
Line 243: Insert a comma following the words „In T. patula,…“
Line 246: Replace the word „in“ by the word „with“ in the sentence „…of more than 90% with respect to their control FW.“
Line 314: Insert a comma after the words „but again,…“
Line 315: Remove the words „of the“ in the sentence „…and it waqs not significant in several cultivars under study…“
Line 344: Insert the word „characteristics“ in the Figure 1 legend „Fig. 1 Relationships between plant growth characteristics, stem length, fresh weight,…“
Line 346: Remove the word „to“ in the sentence „Relationships between plant growth characteristics,…and anti-oxidants…“
Line 355: Fig. 2 legend: Insert a comma following the word „Additionally,…“
Discussion, line 376: Remove a comma in the sentence „T. patula and T. tenuifolia lost more than double of their FW under stress,…“
Line 404: Insert the word „to“ following the word „lead“ in the sentence „Abiotic stresses, including drought, lead to the degradation of photosynthetic pigments…“
Line 423: Remove a comma between the words „species“ and „highlight“ in the sentence „Studies on congeneric wild species or on different cultivars of the same species highlight that…“
Line 426: Insert a comma between the words „here“ and „there“ in the sentence „In all cultivars analysed here, there was a significant increase in Pro,…“
Line 450: Insert a comma before and after the word „especially“ in the sentence „…and, especially, a subgroup of them,…“
Line 454: Remove „the“ before the words „oxidative damage“ in the sentence „…their accumulation in plants can reduce oxidative damage induced by different abiotic stresses,…“
Conclusions, Line 477: Remove „the“ in the sentence „Among osmolytes,…“

·

Basic reporting

OK

Experimental design

OK

Validity of the findings

OK

Additional comments

The authors have made all the changes suggested. Therefore, the manuscript should be accepted for publicción in PeerJ.